# Time-Series analysis of short-term exposure to air pollutants and daily hospital admissions for stroke in Tabriz, Iran

**Shahryar Razzaghi**[1], **Saeid Mousavi**[2]\*, **Mehran Jaberinezhad**[3], **Ali Farshbaf Khalili**[1], **Seyed Mahdi Banan Khojasteh**[4]\*

1 Social Determinants of Health Research Center, Health Management and Safety Promotion Research Institute, Tabriz University of Medical Sciences, Tabriz, Iran, 2 Department of Statistics and Epidemiology, Faculty of Health, Tabriz University of Medical Sciences, Tabriz, Iran, 3 Clinical Research Development Unit of Tabriz Valiasr Hospital, Tabriz University of Medical Sciences, Tabriz, Iran, 4 Department of Animal Biology, Faculty of Natural Sciences, University of Tabriz, Tabriz, Iran

\* Smbanan@tabrizu.ac.ir (SMBK); musavi.stat@gmail.com (SM)

**Data Availability Statement:** All relevant data are within the manuscript and its Supporting information files.

## Abstract

### Background

Air pollution is considered one of the risk factors for stroke prevalence in the long term and incidence in the short term. Tabriz is one of the most important industrial cities in Iran. Hence, air pollution has always been one of the main concerns in environmental health in the region.

### Method

The patient data were retrieved from electronic health records of the primary tertiary hospital of the city (Imam Reza Hospital). Air pollution data was obtained from the Environmental Protection Agency and is generated by 8 sensor stations spread across the city. Average daily values were calculated for CO, NO, NO, $NO_x$, $O_3$, $SO_2$, PM2.5, and PM10 from hourly measurement data. Autoregressive integrated moving average (ARIMA-X) model with 3 lag days was developed to assess the correlation.

### Results

Air pollutants and hospital admission data were collected for 1821 day and includes 4865 stroke cases. our analysis showed no statistically significant association between the daily concentrations of CO (p = 0.41), $NO_x$ (p = 0.96), O3 (p = 0.65), SO2 (p = 0.91), PM2.5 (p = 0.44), and PM10 (p = 0.36). Only the binary COVID variable which was used to distinguish between COVID-19 era and other days, was significant (p value = 0.042). The goodness of fit measures, Root Mean Squared Error (RMSE), and Median Absolute Error (MAE) were 1.81 and 1.19, respectively.

### Conclusion

In contrast to previous reports on the subject, we did not find any pollutant significantly associated with an increased number of stroke patients.

**Funding:** The author(s) received no specific funding for this work.

**Competing interests:** The authors have declared that no competing interests exist.

## Introduction

The world is currently dealing with an epidemic of strokes, which is one of the most significant challenges confronting public health today [1]. Stroke is the second leading cause of mortality and disability which was responsible for 12.2 million incident cases, 101 million prevalent cases, 143 million stroke-related disability-adjusted life years (DALYs) lost, and 6.55 million casualties in 2019 [2]. Furthermore, 87 percent of both stroke-related mortality and DALYs occur in low- and middle-income countries and the incidence of stroke is increasing in these regions [3, 4]. Stroke rose from the fourth cause of mortality in the Middle East and North Africa (MENA) in 1990 to the second cause of mortality in 2019 [5]. Stroke risk factors are classified into biological, lifestyle, genetic, and environmental factors [6–8]. With industrialization and globalization, the significance of new environmental factors, such as various forms of pollution, is becoming more apparent, and environmental factors play a greater role in the development of stroke [9]. Air pollution is the most important environmental risk factor and it rates among modifiable disease risk factors ahead of other major modifiable risk factors, including low physical activity, high sodium intake, and high cholesterol [10, 11].

Air pollution is defined as the presence of any biological, physical, or chemical component that contaminates the outdoor and indoor environments and alters the biochemical makeup of the atmosphere [12]. According to the Global Burden of Disease, air pollution is a major public health concern that contributed to 8.79 million premature deaths worldwide in 2019 [13]. Low and medium-income countries (LMICs) account for 89% of deaths attributed to ambient air pollution, and air pollution in LMICs is anticipated to grow considerably over the coming decades due to increased industrialization [14, 15]. It is also estimated that 12.8% of all deaths in North Africa and Middle east region is attributable to air pollution [16]. Small particulate matter (PM) and gaseous pollutants, including ozone ($O_3$), sulfur dioxide ($SO_2$), nitrogen dioxide ($NO_2$), and carbon monoxide (CO), are the main air pollutants [17].

The impact of air pollution on cerebrovascular accidents can be assessed in the short and long term [18]. Studies on the relationship between air pollution exposure and stroke risk generally separate the risks associated with long-term (months to years) [10, 19, 20] and short-term exposures (over the course of hours to days) [21–25]. Stroke incidence, stroke hospitalization, and stroke mortality are the three indicators used in the studies to evaluate the relationship between air pollution and stroke. The most recent umbrella review and meta-analysis studies in this field indicate a significant association between six major air pollutants (PM2.5, PM10, $NO_2$, $SO_2$, $O_3$, CO) and stroke hospital admission, stroke incidence, and stroke mortality [15, 26]. Nevertheless, the majority of the original studies evaluated in these reviews were conducted in Europe [19, 20, 23–25], North America [10], and East Asia [21], with only one study conducted in the Middle East [22] in 2012. Recent studies of air pollution and health outcomes in Iran are report from Ahvaz city with considerable focus on cardiovascular disease [27, 28]. Considering the paucity of evidence regarding the association between air pollution and stroke in LMICs in the Middle East and the upward trend of air pollution and population aging in these nations, it is deemed necessary to conduct a study in this region. This study addresses this gap by analyzing data from Tabriz, Iran, over a five-year period, utilizing a comprehensive dataset of air pollution levels and hospital admissions and a robust statistical model.

## Method

In this study entitled "Time-Series analysis of short-term exposure to air pollutants and daily hospital admissions for stroke in Tabriz, Iran" we aim to assess the effects of short-term exposure to air pollutants and incidence of stroke.

## Stroke incidence data

Tabriz is a populated industrial city in the northwest of Iran in a mountainous region. The population of Tabriz city in 2022 has reached 1,643,960 (Tabriz County has a population of 1,726,293) with a population density of 7780 people per square kilometer which ranks as the fifth most populated city in Iran. Tabriz experiences a continental climate with low humidity, characterized by hot summers and cold winters.

This study has gathered data pertaining to stroke patients, meticulously retrieved from the Health Information System (HIS) at Imam Reza Hospital, located in Tabriz, Iran. The outcome variable is hospital admissions with a definitive diagnosis. The data collection period spans from March 21, 2018, to March 15, 2023 (1821 days) which includes 4865 cases included in the study. Imam Reza Hospital holds a pivotal role as a tertiary referral center in the Tabriz healthcare landscape, where all patients exhibiting clinical symptoms suggestive of stroke are consistently directed by the Emergency Medical Services (EMS) for evaluation and care. Due to the fibrinolysis treatment only being offered in Imam Reza hospital, only patients within the golden time window are admitted to the neurology ward in this hospital and the remaining are sent to another hospital (Razi) after a confirmed diagnosis by a neurologist. Hence, our stroke patients all receive admission to Imam Reza Hospital (but are not necessarily hospitalized in Imam Reza). Within the HIS system, the diagnosis of stroke adheres to a rigorous protocol, predominantly relying on the final diagnostic assessments rooted in the International Classification of Diseases, Tenth Revision (ICD-10). We specifically enrolled patients whose definitive diagnosis indicated All types of strokes, with the exception of cases involving subarachnoid hemorrhage. The inclusion criteria for this category were defined by diagnostic ICD 10 codes falling within I61 to I64.

## Exposure data

The concentration of air pollutants in Tabriz has been measured by eight air pollution monitoring sites which are strategically located to provide the best possible coverage of the city. These stations take hourly measurements of $O_3$, $CO$, $SO_2$, $NO_2$, $NO$, $NO_x$, and PM2.5 and PM10 particle concentrations. However, it should be noted that different stations have different sensors, and on average, at any given time, four of these eight stations measure $O_3$, three of them measure $CO$, three of them measure $SO_2$, four of them measure $NO$, $NO_2$, and $NO_x$, and five of them measure PM2.5 and PM10. Average daily concentrations of different pollutants were calculated based on these hourly data points. The data for the average daily temperature and relative humidity were obtained from the city's meteorological office.

## Ethical considerations

The present work was approved by the committee on biomedical ethics at the University of Tabriz with the following registration code: IR.TABRIZU.REC.1403.007.

## Statistical analysis

Statistical analysis was performed using Python 3.11 and the statsmodels package (version 0.14.1). Descriptive statistics were calculated for stroke incidence and covariates. In order to find a suitable model for time series data Augmented Dickey–Fuller (ADF) test for stationary was done. Besides, autocorrelation function (ACF) and partial autocorrelation function (PACF) was plotted for time series data to find a suitable model for the data. The analysis included the COVID-19 pandemic period as a binary confounder (March 2020 to March 2022). The significance level for all statistical tests was set at 0.05.

**Table 1. Basic characteristics of stroke admissions, meteorological data, and exposure data.**

|  | Mean | SD | Min | Q1 | Q2 | Q3 | Max |
|---|---|---|---|---|---|---|---|
| Patients (#) | 2.67 | 2.03 | 0 | 1 | 2 | 4 | 13 |
| Temperature (c) | 13 | 9.9 | -10.1 | 4.7 | 13.3 | 21.9 | 33.1 |
| Humidity (%) | 51.53 | 19.09 | 12.87 | 36.25 | 50.75 | 66 | 99.25 |
| CO (ppm) | 1.98 | 0.68 | 0.53 | 1.45 | 1.86 | 2.49 | 6.47 |
| NO (ppb) | 21.22 | 7.68 | 6.06 | 15.93 | 20.08 | 25.02 | 58.33 |
| NO$_2$ (ppb) | 35.23 | 18.21 | 4.75 | 22.39 | 30.96 | 44.57 | 111.38 |
| NO$_x$ (ppb) | 56.30 | 24.19 | 10.86 | 39.36 | 51.61 | 68.53 | 159.66 |
| O$_3$ (ppb) | 19.81 | 8.83 | 2.74 | 12.28 | 19.48 | 27.42 | 41.27 |
| SO$_2$ (ppb) | 6.52 | 5.20 | 0.71 | 3.16 | 5.04 | 8.07 | 54.34 |
| PM 2.5 (µg/m3) | 19.85 | 11.77 | 1.33 | 12.59 | 16.84 | 23.43 | 188.66 |
| PM 10 (µg/m3) | 46.41 | 28.27 | 7.77 | 30.47 | 41.66 | 56.33 | 63.70 |

## Results

Table 1 presents the characteristics of data for Tabriz from March 2018 to March 2023, including the mean, standard deviation, maximum, minimum, and interquartile range (IQR) for study pollutants, metrological variables, and daily stroke admissions. During the study period, 4865 stroke patients were admitted to Tabriz Imam Reza Hospital and the average number of stroke admissions per day was 2.67 ± 2.03.

Fig 1 shows the variation in the daily number of stroke admissions. As we can see there is a visible reduction in the number of cases in the COVID-19 era which proved to be statistically significant in the model (p value = 0.042).

ACF and PACF were plotted for the series (Figs 2 and 3). Non-decaying ACF shows an autocorrelation even in higher lags of the series which could be as a result of trend in the data. Although, the ADF test result indicate a stationary series (p-value<0.05). Also, PACF decays after 3 lags which indicates a moving average degree in the series.

To manage autocorrelation, trend and moving average in the series *ARIMA(p,d,q)* model would be fitted to data. Which p is the order of autocorrelation, d is the order of differencing and p is the order of moving average. Because of non-decaying ACF, the order of autocorrelation was not clear, therefore ARIMA model with different values for p and q was fitted for the data to find the appropriate model. The results are reported in Table 2, in all the models d is set to 1. Model with the lowest Mean Squared Error (MSE) and Akaike Information Criterion (AIC) was selected. ARIMA(3,1,3), demonstrated the lowest MSE and AIC.

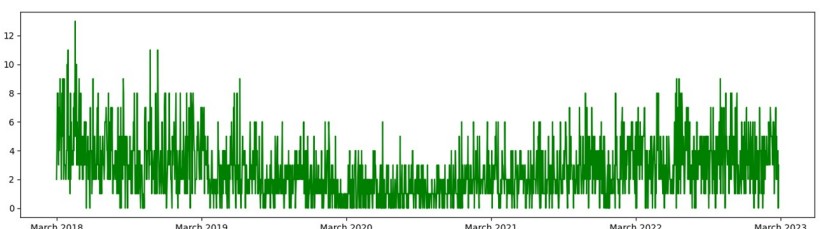

**Fig 1. Number of patients admitted due to stroke in the study period.**

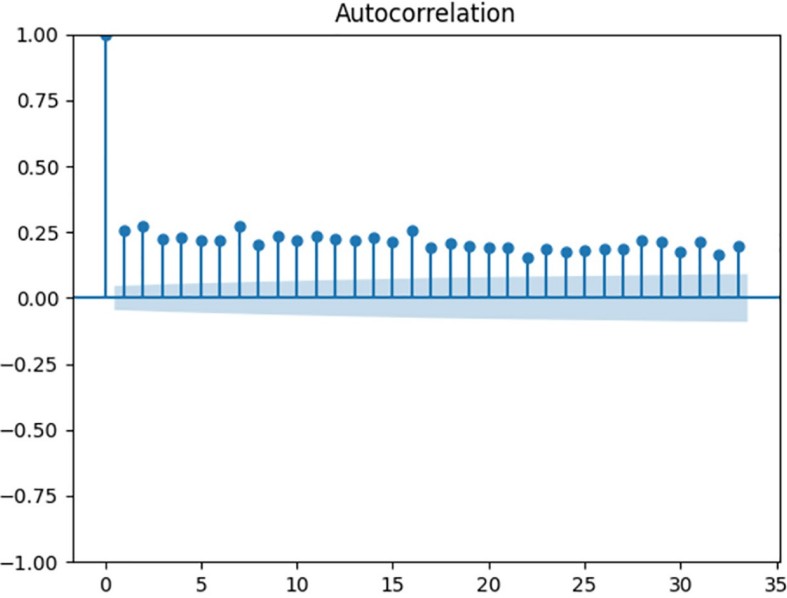

**Fig 2. ACF function for daily stroke incidence data.**

Table 3 shows the results of ARIMA-X (3,1,3) analysis with predictors. The only independent variable that was shown to be significantly associated with number of daily stroke cases was binary covid variable (p value = 0.042). For the ARIMA-X model, the Ljung-Box test shows that the errors are white noise (Q = 0.06, P = 0.80) and the variances are homogenous (H = 0.86, P = 0.07). Also, the goodness of fit measures, Root Mean Squared Error (RMSE), and Median Absolute Error (MAE) were 1.81 and 1.19, respectively.

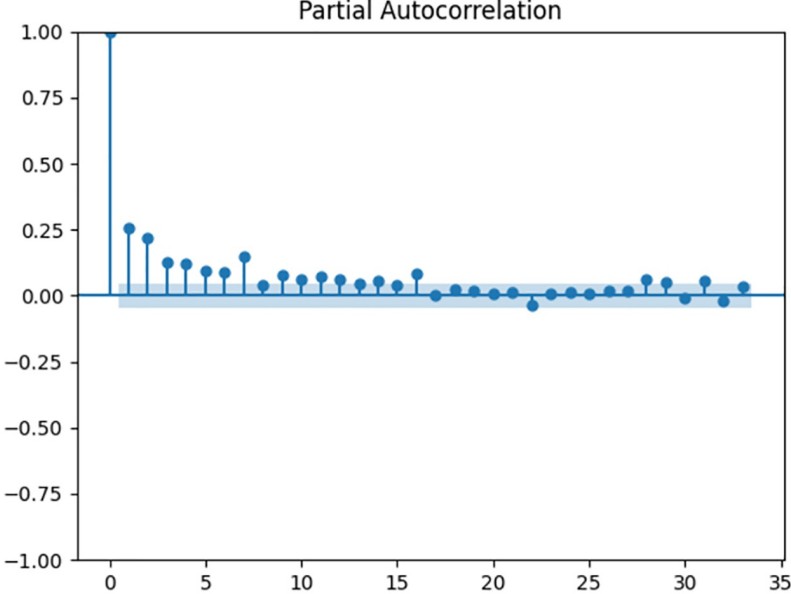

**Fig 3. PACF function for daily stroke incidence data.**

**Table 2. Mean squared error and AIC for different p and q in ARIMA(p,1,q).**

| p | q | MSE | AIC |
|---|---|-----|-----|
| 0 | 0 | 6.0925 | 8452.1989 |
| 0 | 1 | 3.3031 | 7335.1935 |
| 0 | 2 | 3.3006 | 7335.6651 |
| 0 | 3 | 3.2976 | 7335.6332 |
| 1 | 0 | 4.5228 | 7910.5273 |
| 1 | 1 | 3.3005 | 7335.5676 |
| 1 | 2 | 3.3031 | 7339.1975 |
| 1 | 3 | 3.2999 | 7339.5056 |
| 2 | 0 | 4.1336 | 7748.3702 |
| 2 | 1 | 3.2948 | 7334.8520 |
| 2 | 2 | 3.2966 | 7337.4445 |
| 2 | 3 | 3.2926 | 7337.4676 |
| 3 | 0 | 3.9115 | 7650.0427 |
| 3 | 1 | 3.2926 | 7335.1692 |
| 3 | 2 | 3.2915 | 7336.6339 |
| 3 | 3 | 3.2907 | 7333.6352 |

With $d = 1$ difference the $y_i$ and $x_i$ is defined as follows:

$$y_i = Y_i - Y_{i-1}$$

$$x_i = X_i - X_{i-1}$$

The ARIMA-X model is:

$$y_i = \beta x_i + \sum_{j=1}^{p} \phi_j y_{i-j} + \varepsilon_i + \sum_{j=1}^{q} \theta_j \varepsilon_{i-j}$$

**Table 3. Result of fitting ARIMA-X (3,1,3) to assess the effect of independent variables on daily stroke occurrence.**

| | Coefficient | std err | z | P-value | 95% CI | |
|---|---|---|---|---|---|---|
| Covid | -0.8748 | 0.43 | -2.033 | 0.042 | -1.718 | -0.031 |
| CO | 0.1053 | 0.128 | 0.824 | 0.41 | -0.145 | 0.356 |
| $NO_x$ | 0.0002 | 0.003 | 0.05 | 0.96 | -0.006 | 0.007 |
| $O_3$ | -0.0051 | 0.011 | -0.454 | 0.65 | -0.027 | 0.017 |
| PM2.5 | -0.0067 | 0.009 | -0.766 | 0.443 | -0.024 | 0.01 |
| PM10 | 0.0026 | 0.003 | 0.914 | 0.361 | -0.003 | 0.008 |
| $SO_2$ | -0.0015 | 0.013 | -0.113 | 0.91 | -0.028 | 0.025 |
| Temp | -0.0093 | 0.01 | -0.904 | 0.366 | -0.029 | 0.011 |
| ar.L1 | -1.1207 | 0.365 | -3.068 | 0.002 | -1.837 | -0.405 |
| ar.L2 | -0.5756 | 0.245 | -2.354 | 0.019 | -1.055 | -0.096 |
| ar.L3 | 0.0364 | 0.035 | 1.041 | 0.298 | -0.032 | 0.105 |
| ma.L1 | 0.1853 | 0.365 | 0.507 | 0.612 | -0.531 | 0.902 |
| ma.L2 | -0.4656 | 0.263 | -1.773 | 0.076 | -0.98 | 0.049 |
| ma.L3 | -0.6324 | 0.233 | -2.718 | 0.007 | -1.089 | -0.176 |

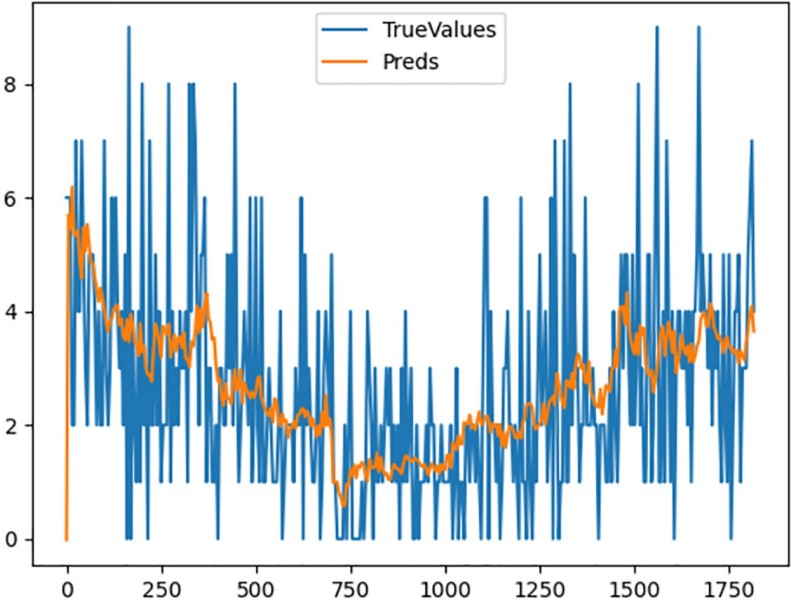

**Fig 4. Expected vs observed number of patients for the ARIMA-X model.**

Which after substituting the estimated parameter in Table 3, the equation of the fitted model is:

$$
\begin{aligned}
y_i &= -0.8748\,\text{Covid} + 0.1053\,\text{CO} + 0.0002\,\text{NOx} - 0.0051\,\text{O3}\\
&\quad - 0.0067\,\text{PM2.5} + 0.0026\,\text{PM10} - 0.0015\,\text{SO2} - 0.0093\,\text{Temp}\\
&\quad - 1.1207\,y_{i-1} - 0.5756\,y_{i-2} + 0.0364\,y_{i-3} + 0.1853\,\varepsilon_{i-1}\\
&\quad - 0.4656\,\varepsilon_{i-2} - 0.6324\,\varepsilon_{i-3} + \varepsilon_i
\end{aligned}
$$

Plotting the expected number of cases calculated based off of this equation against observed data yields Fig 4.

## Discussion

In this article, we investigated the correlation of daily stroke cases in Tabriz and measured the level of different pollutants in the region. And our results suggest an interesting view. Despite a considerable level of air pollution in the city, none of the individual pollutants were significantly associated with number of daily stroke admissions. This discrepancy highlights the complexity of the relationship between air pollution and health outcomes and suggests that regional factors, such as local pollution sources, climate, and population susceptibility, may play a crucial role. These results underscore the importance of conducting similar studies in varied settings to develop a more comprehensive understanding of the health impacts of air pollution globally.

Previous studies have reported a myriad of pollutants that affect this. The most recent meta-analysis study on this topic revealed that while the incidence of stroke increases with increases in PM2.5, $SO_2$, and $NO_2$ pollution concentrations, there was no significant correlation found between the incidence of stroke and increases in $O_3$, CO, and PM10 pollution

concentrations [26]. Our results also concur with non-significance of $O_3$, CO, and PM10 in this subject. However, we also did not observe a significance in other pollutants as well.

In a recent time-series study in China containing 67 078 cases, a direct relationship has been observed between the increase in the concentration of PM2.5, $NO_2$, and $O_3$ pollutants and the increase in the incidence of all types of strokes with 0–3 Lag on [29]. In another time-series analysis in Ireland, it was found that a short-term rise in the concentrations of PM2.5, PM10, $NO_2$, and $SO_2$ pollutants in winter correlates with hospitalization for all strokes in a large urban area [25]. According to the findings of a time-stratified case-crossover study conducted in China, PM2.5 may increase ischemic stroke hospital admissions while $NO_2$ may decrease them, while the effects of $SO_2$ and $O_3$ are statistically insignificant [30]. In an extensive study of 335,248 stroke hospitalizations that was conducted in 97 Japanese cities, it was observed that higher $NO_2$ and photochemical oxidants ($O_x$) concentrations correlate with an increased risk of hospitalization due to ischemic stroke [31]. In the most recent research, conducted in 2022 in China on 824,808 ischemic stroke patients, it was discovered a statistically significant correlation between increasing the concentration of $O_3$ and the risk of ischemic stroke the results were not significant for other pollutants [32].

CO is generated through incomplete combustion of carbon-containing fuels such as gasoline, natural gas, and wood. Motor vehicles, industrial processes, and residential heating systems are common sources of carbon monoxide emissions [33, 34]. The role of CO in cerebrovascular disease remains controversial. As there is evidence of its detrimental effects and increase in risk of stroke [35], and at the same time, endogenous CO helps prevent neural injury in stroke through the regulation of inflammation [36]. The recent meta-analysis did not find evidence of its effect as a pollutant to be statistically significant in incidence or mortality [26].

$O_3$ is a secondary gaseous pollutant that is produced when gaseous precursors, such as $NO_x$ or volatile organic molecules, are exposed to sunlight and undergo a photochemical reaction [37]. One study in southwestern Iran suggested that 12% of all hospital admissions were attributable to Ozone [38]. Studies show that acute exposure to ambient $O_3$ was significantly associated with an increase in blood pressure, angiotensin-converting enzyme (ACE), endothelin-1 (ET-1), and lipid metabolism [39]. Other mechanisms involved in this process are atherogenesis and mitochondrial damage which can lead to vascular dysfunction [40].

$SO_2$ is produced through fossil fuel combustion in power plants, and it is a more prevalent pollutant in LMICs than in high-income countries (HICs) [33]. $SO_2$ has been suggested as an important pollutant in regards to stroke hospital admission [26, 41]. Animal studies on the effects of $SO_2$ have suggested that $SO_2$ inhalation increased ET-1 expression and elevated the levels of proinflammatory enzymes in a concentration-dependent manner in rat cortex [42].

Oxides of nitrogen ($NO_x$) are made when fossil fuels are burned at high temperatures, coming from motor cars and industrial plants. $NO_2$ is a secondary pollutant that forms promptly when oxygen reacts with $NO_x$ [43]. Ambient $NO_x$ inhalation has been implicated in leading to oxidative stress, neuroinflammation, and platelet activation and aggregation. Which in turn would instigate thrombotic events [44]. endothelial nitric oxide synthase (eNOS), cyclooxygenase-2 (COX-2), and intercellular adhesion molecule 1 (ICAM-1) have been cited to be involved in the process [45].

Particulate matter can have a wide range of chemical and physical properties, and emissions can vary greatly between places and over time. PM2.5 is directly produced in urban areas by combustion processes in motor vehicles, industry, and energy production, as well as domestic heating, whereas PM10 mainly consists of crustal material, sea salt, and biological materials [15, 34]. Dust storms, which are becoming prevalent in the region, contribute to PM10 levels which correlate with cardiovascular health outcomes [46]. Systemic oxidative stress and

inflammation have been cited as the main mechanisms of PM2.5 [47]; this is implicated to be propagated by reactive oxygen species which lead to vascular endothelial dysfunction and platelet activation [48].

The inconsistency of the results on the associations between air pollution and stroke might be attributable to variations in air pollution levels, outcome definitions, weather conditions, population susceptibility, and sociodemographic characteristics across studies. And given the plethora of physiologic and pathologic processes affected by different pollutants, the exact observed effect might be non-linear and dependent on the individual risk factors. It must also be noted that lack of correlation in a short time frame does not negate the impact of long-term exposure to air pollutants; considering that 27.1% of the disability adjusted life years (DALYs) due to stroke in our region is attributable to ambient particulate matter pollution [49].

The impact of our research is that it allows some prediction on the number of patients; which in turn may allow healthcare staff to better prepare for incoming patients. This would also help better rationalize healthcare resources and may improve the utilization of such resources. However, it is essential for policymakers and healthcare practitioners to consider regional and local factors when developing strategies to mitigate the health impacts of air pollution. While our results did not show a significant correlation between air pollution and stroke admissions in Tabriz, it is crucial to continue monitoring air quality and health outcomes as a precaution. Policymakers should prioritize enhancing air quality monitoring systems, promoting public awareness about air pollution, and implementing targeted interventions to reduce emissions [50]. Additionally, collaborative efforts between health departments, environmental agencies, and community organizations are necessary to address the multifaceted nature of air pollution and its health impacts [51, 52].

## Limitations

Our study was limited by the resolution of our pollution data and incidence data. We have only been able to cover the area of Tabriz in terms of pollution data. Our pollution data comes from only 8 sensors across the greater Tabriz area and not all of these stations record all the pollutants. Furthermore, the ambient pollution level has been considered as a proxy to the individual exposure. Our incidence data is based on the patients referred to Imam Reza Hospital; which despite being the largest hospital in the region and the point of referral for all stroke cases does not cover 100% of cases in the region. We did not study age since our daily data points were scarce. Similarly, we could not include preexisting risk factors and disease subtypes. Given the discussed limitations, generalizability of our study is somewhat limited; our results may not be directly extrapolated to other regions or populations. Differences in environmental conditions, healthcare infrastructure, socio-economic factors, and population demographics limit the applicability of studies of this type.

One major possible bias in our study is Berkson's bias; as the included population is only composed of stroke patients that are admitted to the hospital. They may differ significantly with general population in predisposing conditions, exposure to different pollutants, and socioeconomic factors.

Finally, we recommend that future research should include community-based studies to contrast and validate our results. Community-based studies can provide a more representative sample of the population and offer a view of the subject with lower risk of bias.

## Conclusion

Our study adds to the body of evidence on correlation between air pollution and stroke incidence. Our findings indicate no significant association between the analyzed pollutants (CO,

NO, NO2, NOx, O3, SO2, PM2.5, PM10) and stroke admissions. These results challenge previous studies and highlight the need for further research in different geographical and socioeconomic settings. Our study emphasize the importance of local context in understanding the health impacts of air pollution and recommends community-based studies in diverse geographical and socioeconomical settings to validate and contrast these findings.

## Supporting information

**S1 Data.**
(CSV)

## Acknowledgments

We would like to thank the Clinical Research Development Unit of Tabriz Valiasr Hospital, Tabriz University of Medical Sciences and University of Tabriz, Iran for their assistance in this research.

## Author Contributions

**Conceptualization:** Shahryar Razzaghi, Seyed Mahdi Banan Khojasteh.

**Data curation:** Shahryar Razzaghi, Saeid Mousavi, Mehran Jaberinezhad, Ali Farshbaf Khalili.

**Formal analysis:** Shahryar Razzaghi, Saeid Mousavi, Mehran Jaberinezhad.

**Investigation:** Seyed Mahdi Banan Khojasteh.

**Project administration:** Saeid Mousavi.

**Resources:** Shahryar Razzaghi.

**Software:** Saeid Mousavi, Mehran Jaberinezhad.

**Validation:** Mehran Jaberinezhad.

**Visualization:** Mehran Jaberinezhad.

**Writing – original draft:** Shahryar Razzaghi, Mehran Jaberinezhad, Ali Farshbaf Khalili, Seyed Mahdi Banan Khojasteh.

**Writing – review & editing:** Shahryar Razzaghi, Saeid Mousavi, Mehran Jaberinezhad, Seyed Mahdi Banan Khojasteh.

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
