## [Decision Letter · Decision Letter 0]

24 Jun 2024

PONE-D-24-18905Air Pollution and daily hospital admissions of stroke patients: A Time-Series Analysis of Exposure in Tabriz, IranPLOS ONE

Dear Dr. Banan Khojasteh,

Thank you for submitting your manuscript to PLOS ONE. After careful consideration, we feel that it has merit but does not fully meet PLOS ONE’s publication criteria as it currently stands. Therefore, we invite you to submit a revised version of the manuscript that addresses the points raised during the review process.

**Major Revisions**

We look forward to receiving your revised manuscript.

Kind regards,

Worradorn Phairuang, Ph.D.

Academic Editor

PLOS ONE

Journal Requirements:

Reviewers' comments:

Reviewer's Responses to Questions

**Comments to the Author**

1. Is the manuscript technically sound, and do the data support the conclusions?

Reviewer #1: No

Reviewer #2: Yes

Reviewer #3: Yes

2. Has the statistical analysis been performed appropriately and rigorously? 

Reviewer #1: No

Reviewer #2: Yes

Reviewer #3: No

3. Have the authors made all data underlying the findings in their manuscript fully available?

Reviewer #1: Yes

Reviewer #2: Yes

Reviewer #3: Yes

4. Is the manuscript presented in an intelligible fashion and written in standard English?

Reviewer #1: No

Reviewer #2: Yes

Reviewer #3: Yes

5. Review Comments to the Author

**Reviewer #1: **- Your manuscript needs some revision.

- The title of this study should be modify.

- Abstract should clearly inform the important findings in the present study.

- The lengthy sentences may be split in to smaller sentence without change of its meaning.

- Your key words seem to be general and should be revised based on MESH.

- Background: The introduction section should be revised.

- Should add some new references published in PLOS ONE.

You can add the following references:

- Borsi SH, Goudarzi G, Sarizadeh G, Dastoorpoor M, Geravandi S, Shahriyari HA, Mohammadi ZA, Mohammadi MJ. Health Endpoint of Exposure to Criteria Air Pollutants in Ambient Air of on a Populated in Ahvaz City, Iran. Frontiers in Public Health. 2022;10.

- Dastoorpoor M, Sekhavatpour Z, Masoumi K, Mohammadi MJ, Aghababaeian H, Khanjani N, Hashemzadeh B, Vahedian M. Air pollution and hospital admissions for cardiovascular diseases in Ahvaz, Iran. Science of the total environment. 2019 Feb 20;652:1318-30.

- Geravandi S, Sicard P, Khaniabadi YO, De Marco A, Ghomeishi A, Goudarzi G, Mahboubi M, Yari AR, Dobaradaran S, Hassani G, Mohammadi MJ. A comparative study of hospital admissions for respiratory diseases during normal and dusty days in Iran. Environmental science and pollution research. 2017 Aug;24:18152-9.

- Effatpanah M, Effatpanah H, Jalali S, Parseh I, Goudarzi G, Barzegar G, Geravandi S, Darabi F, Ghasemian N, Mohammadi MJ. Hospital admission of exposure to air pollution in Ahvaz megacity during 2010–2013. Clinical epidemiology and global health. 2020 Jun 1;8(2):550-6.

- Shahriyari HA, Nikmanesh Y, Jalali S, Tahery N, Zhiani Fard A, Hatamzadeh N, Zarea K, Cheraghi M, Mohammadi MJ. Air pollution and human health risks: mechanisms and clinical manifestations of cardiovascular and respiratory diseases. Toxin Reviews. 2022 Apr 3;41(2):606-17.

- Seihei N, Farhadi M, Takdastan A, Asban P, Kiani F, Mohammadi MJ. Short-term and long-term effects of exposure to PM10. Clinical Epidemiology and Global Health. 2024 May 1;27:101611.

- Abbasi-Kangevari M, Malekpour MR, Masinaei M, Moghaddam SS, Ghamari SH, Abbasi-Kangevari Z, Rezaei N, Rezaei N, Mokdad AH, Naghavi M, Larijani B. Effect of air pollution on disease burden, mortality, and life expectancy in North Africa and the Middle East: a systematic analysis for the Global Burden of Disease Study 2019. The Lancet Planetary Health. 2023 May 1;7(5):e358-69.

- Nikmanesh Y, Mohammadi MJ, Yousefi H, Mansourimoghadam S, Taherian M. The effect of long-term exposure to toxic air pollutants on the increased risk of malignant brain tumors. Reviews on Environmental Health. 2022 Jun 28.

- Hormati M, Mohammadi MJ, Iswanto AH, Mansourimoghadam S, Taifi A, Maleki H, Mustafa YF, Dehaghi BF, Afra A, Taherian M, Kiani F. Consequences and health effects of toxic air pollutants emission by industries. Journal of Air Pollution and Health. 2022 Mar 29;7(1):95-108.

- Yari AR, Goudarzi G, Geravandi S, Dobaradaran S, Yousefi F, Idani E, Jamshidi F, Shirali S, Khishdost M, Mohammadi MJ. Study of ground-level ozone and its health risk assessment in residents in Ahvaz City, Iran during 2013. Toxin reviews. 2016 Oct 1;35(3-4):201-6.

- Materials and Methods: The name of study should be brought in methods section.

- Materials and Methods: please add the time duration of study.

- Materials and Methods: Please describe how the location of sampling was selected, in details.

- Materials and Methods: Statistical analysis of sample data should be modified. This section is unclear.

- Result: The results section should be modified.

- Result: Carefully check that all Tables.

- Result: Please define the abbreviation.

- Discussion: The discussion part should modify.

- Discussion: Refer to more updated articles on similar studies in the discussion section and also reference list.

- Discussion: Please highlight your study's strengths and limitations.

- Discussion: Suggest adding a paragraph on directions for future research, practice and policy.

- Conclusions should be short with important observations.

**Reviewer #2:** Comment to authors

I would like to congratulate on the effort in writing the article: "Air Pollution and Daily Hospital Admissions of Stroke Patients: A time-series analysis of exposure in Tabriz, Iran".

I must say that this article is very interesting and will contribute to know the usefulness of time series studies.

I left you some observations to improve the quality of the information you tray to communicate:

Title: I think you need to include in the title: Short-term associations

Short-term associations of air pollution and daily hospital admissions in stroke patients: A time-series analysis of exposure in Tabriz, Iran

Introduction:

The authors say:

“The most recent umbrella review and meta analysis studies in this field indicate a significant association between six major air pollutants and stroke incidence, stroke hospitalization, and stroke mortality[16, 25].”

Please specify which six major air pollutants.

Methods

1.- Please state the number of cases included.

2.- When you use ICD 10 you say:

The inclusion criteria for this category were defined by diagnostic ICD 10 codes falling within 161 to 164.

Please check, because I think the code is I61 to I64. With (I) before nor number 1 before.

Please specify why you don't analyse different types of stroke.

Exposure data:

Please specify outcome variable: I assume that the outcome variable is hospital admissions of stroke patients, but you don't specify this variable in the text, please clarify e.g. whether the outcome variable is admissions or hospitalised patients and why you don't use hospitalised patients.

In the methods, you must specify the reason why you are using the ARIMA model and not another model, or classical analysis of time series studies and the reason why you are using the ARIMA model with this description as (3,1,3) each parameter has a reason, please specify.

A cross correlation graph may be useful.

Limitations:

It is necessary to include in the limitations the possibility that you study is affected for Berkson byas, because it is not a community study, please review about this byas and include this information in the limitations part.

Please include in the limitations that your study can't be extrapolated to other realities.

You may recommend that this study should at least be done in the community to contrast the results.

**Reviewer #3: **This study lack of novelty. novelty of the study should be highlighted.

Title needs to be rewritten.

The implemented model id ARIMA or ARIMA-X model.

Necessary equation and flowchart of the model should be included.

Detailed methodology should be provided.

6. PLOS authors have the option to publish the peer review history of their article (what does this mean?). If published, this will include your full peer review and any attached files.

Reviewer #1: No

Reviewer #2: **Yes: **DANTE ROGER CULQUI LEVANO

Reviewer #3: No

---

## [Author Response · Author response to Decision Letter 0]

7 Aug 2024

Dear Prof. Phairuang,

I hope this letter finds you well. I am writing to address a concern regarding the peer review comments for our manuscript titled " Time-Series analysis of short-term exposure to air pollutants and daily hospital admissions for stroke in Tabriz, Iran "

One of the reviewers has suggested that we cite ten of their own publications in our manuscript. While we appreciate the reviewer’s efforts and insights, we believe that this request raises ethical concerns, as many of the suggested citations are not directly relevant to our study's subject matter.

Our manuscript focuses on the impact of air pollutants on the incidence of stroke in the short term, but most of the studies suggested by reviewer #1 are on the issue of the impact of air pollutants on the incidence of cardiopulmonary diseases. We are committed to maintaining the integrity and scientific rigor of our work by including references that genuinely contribute to and support our findings. Therefore, we find it inappropriate to incorporate citations that do not align with the scope and relevance of our research.

Furthermore, the reviewer recommended that we use newer references, yet proceeded to suggest research conducted between 2010 and 2013. This contradiction adds to our concern about the appropriateness of the suggested citations.

Additionally, we have noted that some of the reviewer's comments were vague and lacked constructive criticism. For instance, the comment "introduction needs to be modified" was provided without any specific explanation or guidance. Such feedback does not help in improving the manuscript and leaves the authors without clear direction on how to address the reviewer's concerns.

We have revised our original manuscript as instructed and incorporated the peer reviewer comments as best as possible. However, we feel compelled to bring this matter to your attention to ensure the integrity of the review process and uphold the standards of the journal.

Thank you for your understanding and consideration. 

Sincerely,

Seyed Mahdi Banan Khojasteh

---

## [Decision Letter · Decision Letter 1]

13 Aug 2024

Time-Series analysis of short-term exposure to air pollutants and daily hospital admissions for stroke in Tabriz, Iran

PONE-D-24-18905R1

Dear Dr. Banan Khojasteh,

We’re pleased to inform you that your manuscript has been judged scientifically suitable for publication and will be formally accepted for publication once it meets all outstanding technical requirements.

Kind regards,

Worradorn Phairuang, Ph.D.

Academic Editor

PLOS ONE

Additional Editor Comments (optional):

Reviewers' comments:

Reviewer's Responses to Questions

**Comments to the Author**

1. If the authors have adequately addressed your comments raised in a previous round of review and you feel that this manuscript is now acceptable for publication, you may indicate that here to bypass the “Comments to the Author” section, enter your conflict of interest statement in the “Confidential to Editor” section, and submit your "Accept" recommendation.

Reviewer #1: All comments have been addressed

Reviewer #2: All comments have been addressed

2. Is the manuscript technically sound, and do the data support the conclusions?

Reviewer #1: No

Reviewer #2: Yes

3. Has the statistical analysis been performed appropriately and rigorously? 

Reviewer #1: Yes

Reviewer #2: Yes

4. Have the authors made all data underlying the findings in their manuscript fully available?

Reviewer #1: No

Reviewer #2: Yes

5. Is the manuscript presented in an intelligible fashion and written in standard English?

Reviewer #1: Yes

Reviewer #2: Yes

6. Review Comments to the Author

Reviewer #1: The revised manuscript has been modified according to comments of reviewers and it can be ACCEPT for publication.

Reviewer #2: After reviewing the latest version of the article entitled: "Time-Series analysis of short-term exposure to air pollutants and daily hospital admissions for stroke in Tabriz, Iran", I believe that this article is suitable for publication.

7. PLOS authors have the option to publish the peer review history of their article (what does this mean?). If published, this will include your full peer review and any attached files.

Reviewer #1: No

Reviewer #2: **Yes: **DANTE ROGER CULQUI LEVANO (https://orcid.org/0000-0003-1570-8012))

---

## [Editor Report · Acceptance letter]

28 Aug 2024

PONE-D-24-18905R1 

PLOS ONE

Dear Dr. Banan Khojasteh, 

I'm pleased to inform you that your manuscript has been deemed suitable for publication in PLOS ONE. Congratulations! Your manuscript is now being handed over to our production team.

Kind regards, 

on behalf of

Dr. Worradorn Phairuang 

Academic Editor

PLOS ONE